# Defective activation and regulation of type I interferon immunity is associated with increasing COVID-19 severity

Host immunity to infection with SARS-CoV-2 is highly variable, dictating diverse clinical outcomes ranging from asymptomatic to severe disease and death. We previously reported reduced type I interferon in severe COVID-19 patients preceded clinical worsening. Further studies identified genetic mutations in *loci* of the TLR3- or TLR7-dependent interferon-I pathways, or neutralizing interferon-I autoantibodies as risk factors for development of COVID-19 pneumonia. Here we show in patient cohorts with different severities of COVID-19, that baseline plasma interferon α measures differ according to the immunoassay used, timing of sampling, the interferon α subtype measured, and the presence of autoantibodies. We also show a consistently reduced induction of interferon-I proteins in hospitalized COVID-19 patients upon immune stimulation, that is not associated with detectable neutralizing autoantibodies against interferon α or interferon ω. Intracellular proteomic analysis shows increased monocyte numbers in hospitalized COVID-19 patients but impaired interferon-I response after stimulation. We confirm this by ex vivo whole blood stimulation with interferon-I which induces transcriptomic responses associated with inflammation in hospitalized COVID-19 patients, that is not seen in controls or non-hospitalized moderate cases. These results may explain the dichotomy of the poor clinical response to interferon-I based treatments in late stage COVID-19, despite the importance of interferon-I in early acute infection and may guide alternative therapeutic strategies.

Type I interferons (IFN-I) are key components of innate anti-viral immune activity and therefore a target for viral interference strategies[1]. We previously reported impaired IFN-I responses in severe COVID-19 patients that preceded clinical worsening[2]. These observations were further supported by studies led by the COVID Human Genetics Effort (www.covidhge.com) which identified mutations in loci that govern Toll-Like Receptor (TLR)3-dependent and interferon regulatory factor (IRF)7-dependent IFN-I immunity[3], or autoantibodies against IFNα, IFNω, or IFNβ[4,5], as major risk factors for the development of severe COVID associated pneumonia[6,7]. Additional genetic studies have identified the TLR7 pathway to also be critical for host immunity to infection with SARS-CoV-2[8]. Despite this key role in early immunity to SARS-CoV-2 infection, the use of exogenous IFN-I as a treatment for COVID-19[9,10] has not improved clinical outcomes. However, in these studies, IFN-I was given late in the disease course and patients were not stratified[9,10]. The dichotomous IFN-I activity likely reflects the two-step nature of IFN-I responses in COVID-19 pathogenesis[11], with the first step characterized by high IFN-I activity required for viral suppression by innate anti-viral immunity. If step 1 is ineffective, for reasons described above, and the virus is not cleared, viral dissemination, hyper-inflammation, and compromised adaptive immunity occur, followed by pneumonia and death in a significant proportion of patients.

✉ e-mail: darragh.duffy@pasteur.fr

Despite the clear importance of IFN-I in early innate immunity against SARS-CoV-2 infection, previous studies have reported increased levels of IFNs[12] and interferon-stimulated genes (ISGs)[13] as biomarkers for mortality. These differences in reporting the precise role of IFN-I may be due to differences in disease kinetics[14,15], patient populations[16], reporting of IFN protein data versus ISG expression[17,18], multiple IFN-I subtypes[19,20], the anatomical site studied[21,22] or even technical differences in the assays employed[23]. Indeed, IFNα protein has been notoriously challenging to measure, leading to the use of ISGs as a proxy readout for IFN signaling. We previously developed a digital ELISA that permits the ultrasensitive detection of all IFNα subtypes[24], or specifically the IFNα2 subtype[25]. Notably, in blood samples from multiple autoimmune cohorts with clinically known IFN-driven pathology, IFNα protein levels were below the limits of detection of conventional ELISA/Luminex assays but were quantifiable using our approach[24]. This highlights the importance of employing sufficiently sensitive and qualified immunoassays when studying type I IFN directly from patient samples. Furthermore, while plasma IFN levels may reflect in vivo anti-viral activity at the moment of patient sampling, it does not necessarily inform on the ability of the patient's cells to respond to a viral encounter. This requires the use of functional immune assays with standardized approaches that minimize technical variability[26], particularly important in heterogenous patient populations.

To better clarify how different IFN-I measures might be used to understand COVID-19 pathogenesis, we compared cases of moderate COVID-19 with patients hospitalized for severe disease across different countries and clinical centers. For baseline responses, we characterized IFNα proteins with highly sensitive assays recognizing either specifically IFNα2 or all 13 alpha subtypes, as well as IFN function, ISGs, and autoantibodies neutralizing IFN-I. In a subgroup of these patients, we stimulated whole blood with relevant viral agonists to further assess the functional capacity of their immune system to respond to an external perturbation. We highlight the importance of using sufficiently sensitive and qualified assays for measuring IFN-I proteins and report defective IFN-I induction that is consistent in all hospitalized COVID-19 patients. Defective IFN-I induction in response to these stimuli was not due to autoantibodies potentially masking the protein, except in one patient. We also demonstrate that ex vivo IFN-I stimulation of blood from hospitalized patients induces a non-canonical inflammatory response, perhaps explaining the poor clinical outcome of IFN-I-based treatments previously reported. Our study highlights the importance of this crucial anti-viral immune response in host protection against SARS-CoV-2 infection, supports strategies for earlier more targeted therapeutic intervention, and highlights the importance of using consistent technical approaches for the investigation of IFN-Is in human cohort studies.

## Results

### Decreased blood IFNα protein, ISGs, and activity in severe and critical COVID-19 patients

We previously reported lower plasma IFNα2 levels in COVID-19 patients with severe and critical disease, when assessed 8–12 days post-symptom onset[2]. A multinomial logistic regression model integrating age and sex (Supplemental Data 1) showed significantly higher levels of plasma IFNα2 in moderate ($P = 0.005$) and severe ($P = 0.03$) disease compared to uninfected controls, but no increase in the critical group ($P = 0.33$) (Fig. 1a). Using the moderate group as the reference, significantly ($P = 0.02$) lower levels were present in the critical group (Fig. 1a). To test if this was also observed for all 13 IFNα subtypes, we applied the multi-IFNα subtypes digital ELISA (measured as the equivalent of IFNα17) and observed a similar pattern, with significantly higher levels of all IFNα subtypes in moderate ($P = 0.002$) and severe ($P = 0.005$), and also a significant increase in critical patients ($P = 0.01$), as compared to non-infected healthy controls. Using the moderate

group as the reference, significantly ($P = 0.005$) lower levels were present in the critical group (Fig. 1b). Due to conflicting reports on IFNα levels in other studies using non-digital assays, we assessed IFNα2 in the same samples using a commercial Luminex assay. Strikingly, results from this assay showed no differences between controls and moderate, severe, and critical COVID-19 patients (Fig. 1c). To test which protein assay best reflected in vivo activity, we correlated measurements from each assay with an ISG score (6 gene score, previously validated and described in the "Methods" section[2]) (Fig. 1d–f), and with IFN activity measured by a functional cytopathic assay previously described[2] (Fig. 1g–i). IFNα2 and multi IFNα subtype proteins measured by digital ELISA showed good positive correlations with both ISG score ($R_s = 0.69$, $R_s = 0.82$) (Fig. 1d, e) and functional activity ($R_s = 0.51$, $R_s = 0.57$) (Fig. 1g, h), in contrast with the Luminex values which did not correlate with either ISG score ($R_s = 0.07$) (Fig. 1f) or functional activity ($R_s = 0.12$) (Fig. 1i). The two digital ELISA measures strongly correlated with each other (Fig. S1a), but neither correlated with the Luminex values (Fig. S1b, c). These collective results highlight the importance of using sensitive and qualified assays for studying IFN-I proteins in human samples. Given the challenges of comparing different patient severities in terms of time post-infection, we also examined whether time–post-symptom onset was associated with IFNα protein levels, but no significant associations were observed for each patient severity with the three separate assays (Fig. S1d–f). In contrast to IFNα, we detected no IFNβ in these plasma samples using a Simoa assay with a limit of detection of 0.6 pg/ml (Fig. S1g).

To validate our results in an independent cohort, and also assess potential differences with non-alpha SARS-CoV-2 viral variants, we applied the three IFNα assays to another cohort of COVID-19 patients (Supplemental Data 2) recruited during a different wave of infection when the delta variant was prominent (December 2020–April 2021). The overall pattern of results was similar with the lowest protein levels observed in critical patients using the digital ELISAs (Fig. 1j, k), and no differences between patient groups using Luminex (Fig. 1l). Multimodal logistic regression models integrating age and sex, using the moderate group as a reference, showed significantly lower IFNα2 ($P = 0.04$) and multi-IFNα subtype ($P = 0.01$) proteins in the critical patient group. Interestingly much greater heterogeneity (5 logs of variability compared to 3) was observed in the IFNα plasma levels in this replication cohort compared to the first cohort (Fig. 1a–c) likely reflecting a more diverse and older patient cohort (median ages; 55 years cohort 1 and 66 years cohort 2, $P < 0.001$) (Supplemental Data 2). As in the first cohort, the two digital ELISA measures strongly correlated with each other (Fig. S1h), but neither correlated with the Luminex values (Fig. S1i, j). These results highlight the challenges in comparing cytokine responses across clinical cohorts with different underlying characteristics and the importance of considering such variables in the interpretation of immune data during active infection.

Given the high variability observed in IFN-I plasma levels, as well as the challenges of obtaining samples from early in infection, we tested whether we could observe significant differences in IFN-I protein levels between COVID-19 patients that were either non-hospitalized or hospitalized based on their requirement for oxygen supplementation. For this analysis, patients were recruited at St James's Hospital (SJH) in Dublin, Ireland as described in the "Methods" section (Supplemental Data 3). Multimodal regression models incorporating age and sex, and using healthy controls as the baseline showed a significant ($P = 0.008$) increase in IFNα2 (Fig. 2a) levels in non-hospitalized, but not hospitalized patients ($P = 0.24$). In contrast, results with the multi-IFNα subtype assay showed a significant increase in all subtypes in both non-hospitalized ($P = 0.04$) and hospitalized patients ($P = 0.03$) (Fig. 2b). Confirming these potential IFNα subtype differences a direct comparison between the two COVID-19 patient groups showed a significant

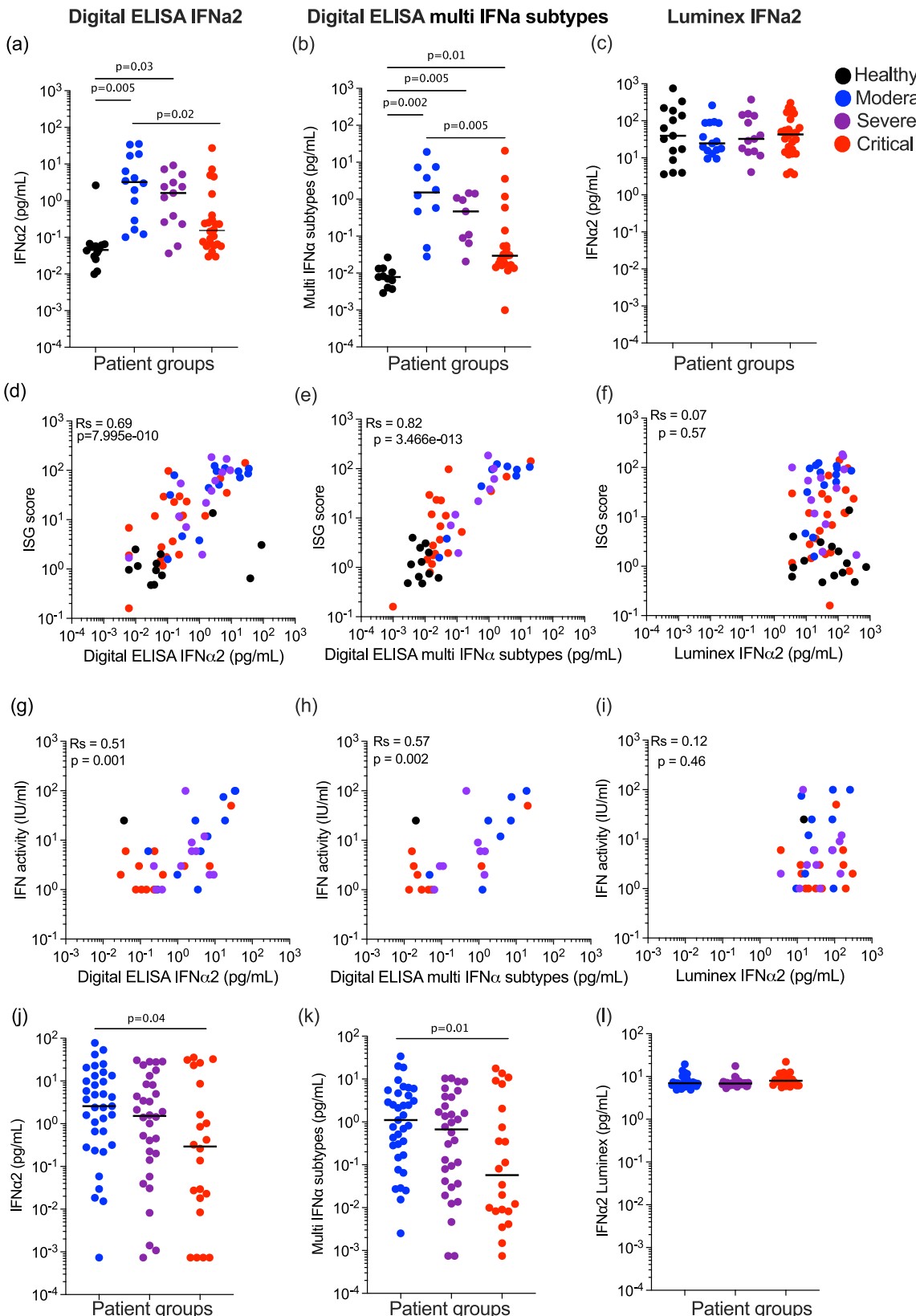

difference ($P = 0.004$) in IFNα2 levels only (Fig. 2a, b). In agreement with the previous results, IFNβ levels were undetectable in the majority of patients, with no differences between the groups and controls (Fig. S2a). An ISG score ($Z$ score of ISGs measured by nanostring, described in the "Methods" section) in paired whole blood samples correlated positively and significantly with both Simoa IFNα measures (IFNα2:

$R_s = 0.36$, $P = 0.005$ Fig. S2b, and multi-IFNα subtype: $R_s = 0.42$, $P = 0.002$, Fig. S2c); but not IFNβ (Fig. S2d).

Given the observed differences in IFNα2, but not all IFNα subtypes, at the plasma protein level, we further explored potential differences in IFNα subtype expression using Nanostring gene expression data on whole blood. This showed low transcriptional levels of all

**Fig. 1 | Plasma IFNα is consistently reduced with increasing severity of Covid-19. a** IFNα2 (left) and **b** multi-IFNα subtypes (=equivalent IFNα17) were measured by Simoa digital ELISA or **c** Luminex in healthy controls (n = 14 donors) and in patients with moderate (n = 15), severe (n = 13) and critical (n = 27) disease from the first Hopital Cochin cohort. Interferon-stimulated gene (ISG) score (6 gene score) correlated with IFNα levels measured by **d** Simoa IFNα2, **e** Simoa multi IFNα subtypes or **f** Luminex; n = 64. IFN activity (IU/mL) correlated with IFNα levels measured by **g** Simoa IFNα2, **h** Simoa multi IFNα subtypes, or **i** Luminex IFNα2; n = 35. **j** IFNα2, **k** multi-IFNα subtypes (=equivalent IFNα17) were measured by Simoa digital Elisa or **l** IFNα2 by Luminex in patients with moderate (n = 35), severe (n = 32) and critical (n = 21) disease from the second Hopital Cochin cohort. Lines indicate median values. *P* values were determined by multimodal regression models incorporating age and sex. *R*s represents the Spearman coefficient. Healthy control = black, moderate COVID-19 patients = blue, severe COVID-19 patients = purple and critical COVID-19 patients = red. *N* = number of individual patients included. Source data are provided as a Source Data file.

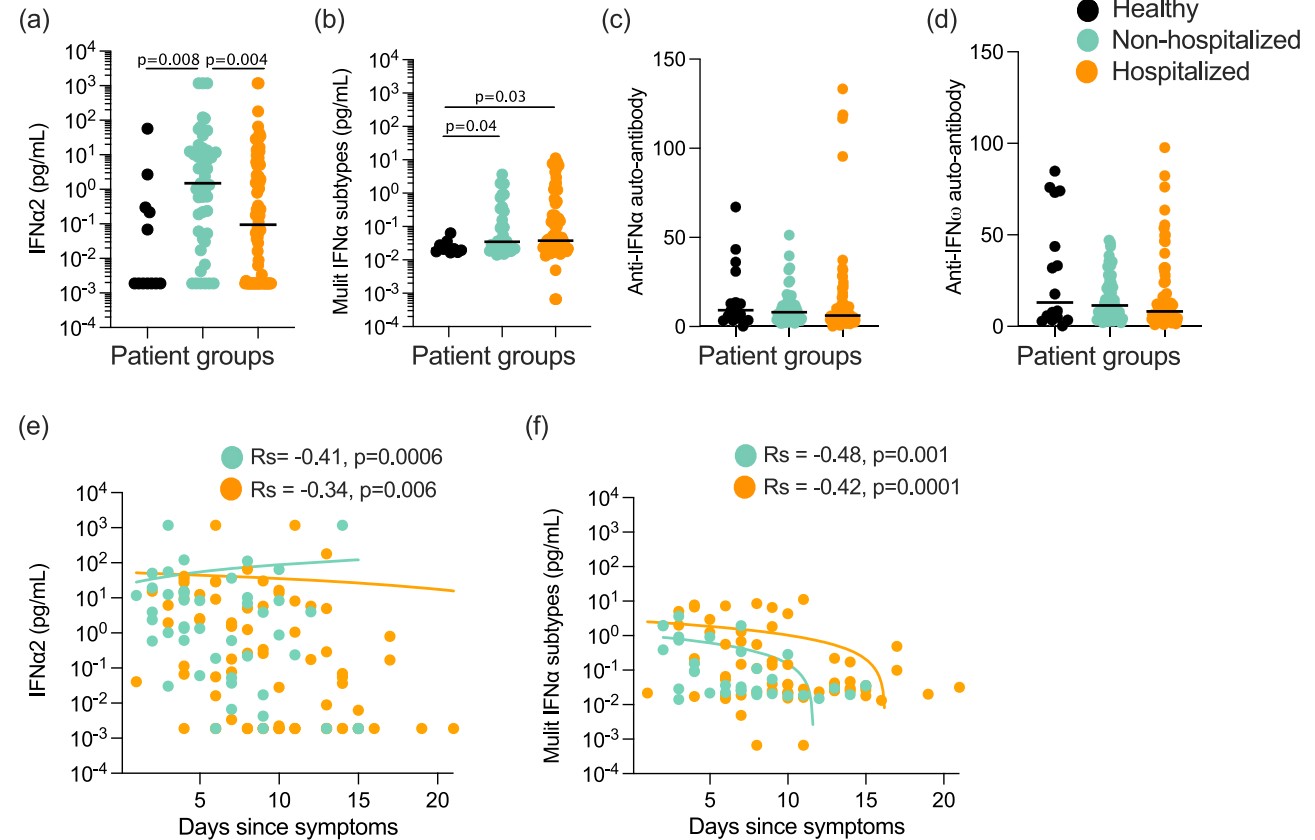

**Fig. 2 | Plasma IFNα is consistently reduced in hospitalized patients independent of neutralizing auto-antibodies. a** IFNα2 and **b** multi-IFNα subtypes (=equivalent IFNα17) measured by digital ELISA Simoa in healthy controls (n = 14 donors), non-hospitalized (n = 51), or hospitalized (n = 85) COVID-19 patients of the St. James Hospital cohort. **c** Anti-IFNα and **d** anti-IFNω auto-antibodies in plasma of these healthy controls and COVID-19 patients. **e** IFNα2 and **f** multi-IFNα subtype cytokine levels as a function of the number of days post symptoms, with regression lines per COVID-19 patient groups shown. P values were determined by multimodal regression models incorporating age and sex. Lines indicate the median values. Healthy control = black, non-hospitalized COVID-19 patients = green, and hospitalized COVID-19 patients = orange. *R*s indicates Spearman correlation, n = number of individual patients included. Source data are provided as a Source Data file.

measured *IFNA* subtypes as expected (Fig. S2e). Despite these low baseline levels, among the 7 *IFNA* subtypes examined we did observe some subtype differences, with notably higher levels of *IFNA6* and *IFNA5* in both COVID-19 patient groups and *IFNA1/13* only in the hospitalized group (Fig. S2e). *IFNA2* was notably no different between all three groups (Fig. S2e).

Given the previously reported importance of autoantibodies as a risk factor for severe COVID-19[4], and their potential to interfere with IFNα protein measurements[25], we quantified anti-IFNα (Fig. 2c) and anti-IFNω (Fig. 2d) autoantibodies by Gyros assay. Among the 126 patients tested, 4 were identified as anti-IFNα autoantibody-positive (>95 was considered positive, based on the distribution in previously published healthy controls[5]) and all were male patients (aged 31, 53, 81, and 85 years old) in the hospitalized group with mixed severities at time of sampling (2 moderate, 1 severe, 1 critical) (Fig. 2c). Interestingly, of these 4 patients, multi IFNα subtype results were

extremely low (Fig. S2g), although 2 had elevated levels with the IFNα2 assay (Fig. S2f). We previously described how recognition of IFNα2 in this specific assay is not always blocked by autoantibodies from all autoantibody-positive individuals[25], likely due to recognition of a non-functional epitope in contrast with the multi IFNα subtypes assay. However, the majority of patients were negative for anti-IFN autoantibodies suggesting that this was not the driver of severe disease in these patients.

We next assessed whether IFNα levels changed with time post-symptoms in either patient group in this cohort. IFNα2 (Fig. 2e) and multi IFNα subtype (Fig. 2f) levels mostly declined with time post-symptoms in both patient groups (non-hospitalized; *R*s = −0.41 and −0.48, and hospitalized; *R*s = −0.34 and −0.42) with no significant differences in the decline (*P* = 0.45, *P* = 0.68 for IFNα2 and multi-IFNα subtypes respectively) between the two patient groups (Fig. 2e, f). IFNβ levels, although lower to begin with, also followed the same decline

with time post-symptoms with no significant difference ($P = 0.08$) between the two patient groups (Fig. S2h).

## Induction of type I interferon is compromised in hospitalized patients

Our results thus far demonstrating lower IFNα plasma levels in severe disease, yet similar kinetics across COVID-19 disease states, suggests that induction of IFN-I in the absence of autoantibodies is critical to understand defective anti-viral immunity in severe COVID-19. To investigate the functional IFN capacity of COVID-19 patients, whole blood from a sub-cohort of patients (Supplemental Data 4) was stimulated with immune agonists relevant for anti-viral immune pathway activation, namely Poly:IC (a synthetic analog of double-stranded RNA and reported TLR3/MDA5 agonist), R848 (a small molecular weight imidazoquinoline compound and TLR7/8 agonist), as well as LPS (Lipopolysaccharide (LPS) synthesized by *E. coli* and a TLR4 agonist), and a Null condition as positive and negative controls, respectively. At 22 hrs, IFNα2 was weakly induced by Poly:IC stimulation, but strongly induced by R848 (Fig. 3a). Measurement of all IFNα subtypes revealed a broader response with the highest levels in non-hospitalized patients (Fig. 3b). Multinomial logistic regression models integrating age and sex, and using the healthy group as the reference, showed significantly higher ($P = 0.01$) multi-IFNα subtype levels in non-hospitalized moderate patients after Poly:IC ($P = 0.01$) and LPS ($P = 0.002$) stimulation, but not in more severe patients (Fig. 3b). More striking was the IFNβ response which was significantly reduced ($P < 0.001$) after Poly:IC stimulation in both COVID-19 groups compared to controls, but also significantly reduced after R848 ($P = 0.005$) and LPS ($P = 0.04$) stimulation in only the hospitalized groups (Fig. 3c).

To assess whether differences in receptor expression could explain these cytokine differences, we examined the relevant TLR gene expression data from whole blood. Expression of *TLR7* was similar in all groups, while expression of *TLR3* was significantly ($P = 0.04$) lower, and expression of *TLR4* ($P = 0.01$) and *TLR8* ($P = 0.003$) was significantly higher, in hospitalized patients. Expression of *IFIH1* was higher in both patient groups, although the effects were modest and the differences are unlikely to explain the differences in cytokine responses (Fig. 3d). To examine whether these immune differences were restricted to IFN-I responses, we measured an additional 42 cytokines in the whole blood stimulations by Luminex (Fig. S3, Supplemental Data 5). This revealed 8 additional cytokines with significant differences in hospitalized patients, mostly after TLR3 stimulation (Fig. S3). However, half of these differential cytokines (CXCL10, IL-12p70, CCL4, and IL-10), which were all lower in hospitalized patients after Poly:IC stimulation, are downstream of IFN-I responses. This further supports the finding of defective IFN-I responses in unfavorable COVID-19 states.

To confirm these observations in an independent cohort of COVID-19 patients with more severe disease, we sampled additional hospitalized patients from two Parisian clinical centers (Supplemental Data 5). Given previous studies indicating age and sex as strong risk factors, we also recruited severe and critical patients with similar ages (medians; 63 and 76 years old, respectively, $P = 0.05$) and sex distribution (40% and 26% female, Fisher test 0.7) (Supplementary Data 5). We also tested lower agonist concentrations, to assess more subtle induction of IFN-I responses and avoid potential over-stimulation, to which acutely infected patients may be more sensitive. In this independent cohort, we again observed a strongly reduced IFNα2 (Fig. 3e–g) and IFNβ (Fig. 3h–j) secreted response to Poly:IC, LPS and R848 stimulation in critical COVID-19 patients in comparison to healthy controls. Severe COVID-19 patients showed an intermediate response. We also included a live viral stimulus (influenza H1N1 PR8 strain), to which the IFN-I (IFNα2 and IFNβ) response was also significantly reduced as the disease severity increased (Fig. 3k, l). Collectively these results show a broadly perturbed type I interferon response to diverse immune stimulation that increases with disease severity.

## Compromised interferon responses at the cellular level with increasing COVID-19 severity

Previous studies have reported multiple cellular and intracellular perturbations in patients with moderate as well as severe COVID-19. To test whether changes in either cell numbers or interferon regulatory transcription factors (IRFs), could explain the severely reduced IFN responses to pattern recognition receptor (PRR) stimulation, we performed multi-parameter intra-cellular flow cytometry on the blood of hospitalized COVID-19 patients (Supplemental Data 5, Fig. S4a, Table S2). In line with previous findings, we observed a significant increase in circulating granulocytes and monocytes, in parallel to a significant decline in T cells and pDCs, in critically infected patients confirming that our cohort exhibited the typical immunological dysregulation associated with increasing severity of COVID-19 (Fig. 4a). To assess IFN-I signaling pathways we measured intracellular phosphorylated IRF3 and IRF7, and intracellular IFNα2 after R848 stimulation (Fig. S4b–d). Analysis of the percentage of positive cells and mean fluorescent intensity (MFI) showed that pIRF7 significantly increased in different monocyte subsets and pDCs of healthy donors after R848 stimulation (Fig. S4b). Changes in percentages and MFI of pIRF3 and intracellular IFNα2 were more modest but were also detectable in monocytes and pDCs after R848 stimulation (Fig. S4c, d).

As our data showed that both pDC and monocyte cell subsets are major regulators of the type I IFN response in human blood in response to R848, we, therefore, focused on these cell types in COVID-19 patients. We observed a significant increase in pIRF7 in monocytes from severe and critical groups (Fig. 4b). However, this was not matched by an increase in intracellular IFNα2, perhaps due to the lack of induction of pIRF3 (Fig. 4c), or the elevated intracellular IFNα2 at the baseline in the absence of stimulation (Fig. 4d) in both severe and critical patients. In pDCs of COVID-19 patients, pIRF7 was also significantly increased after R848 stimulation (Fig. 4e), but in contrast to the monocytes this was matched by an increase in intracellular IFNα in severe, but not in critical patients (Fig. 4g). Additional correlation analysis between cytometry measured intracellular IFNα, and digital ELISA measured plasma IFNα, showed in the absence of stimulation, an association between monocytes and plasma IFNα levels (Fig. 4h). Following R848 stimulation, both pDCs and monocytes showed an association with secreted IFNα (Fig. 4i), although the percentage of IFNα+ cells was lower in critical patients compared to severe and healthy controls (Fig. 4d and g). Collectively these results indicate a perturbation of IFN signaling networks in the elevated numbers of circulating monocytes in all hospitalized COVID-19 patients. pDCs demonstrate a quantitative decrease in disease states but retain functional responsiveness in severe disease. This pDC functional response is lost in the critical disease.

## Induced gene expression changes identify consistently perturbed myeloid-associated pathways

To further explore potential reasons behind the perturbed IFN-I responses in hospitalized patients, we examined gene expression differences after immune stimulation between hospitalized and non-hospitalized patients in the St. James cohort (Supplemental Data 4). To do this, we applied UMAP (Uniform Manifold Approximation and Projection for Dimension Reduction) to 800 immunology and host response-related genes measured by Nanostring in each condition (Null, PolyI:C, LPS, and R848) (Fig. 5a). This revealed clustering of the hospitalized patients and healthy controls, with the non-hospitalized patients spread between the two groups in all conditions including the unstimulated control. This suggests that induced immune responses are already perturbed at baseline in hospitalized COVID-19 patients.

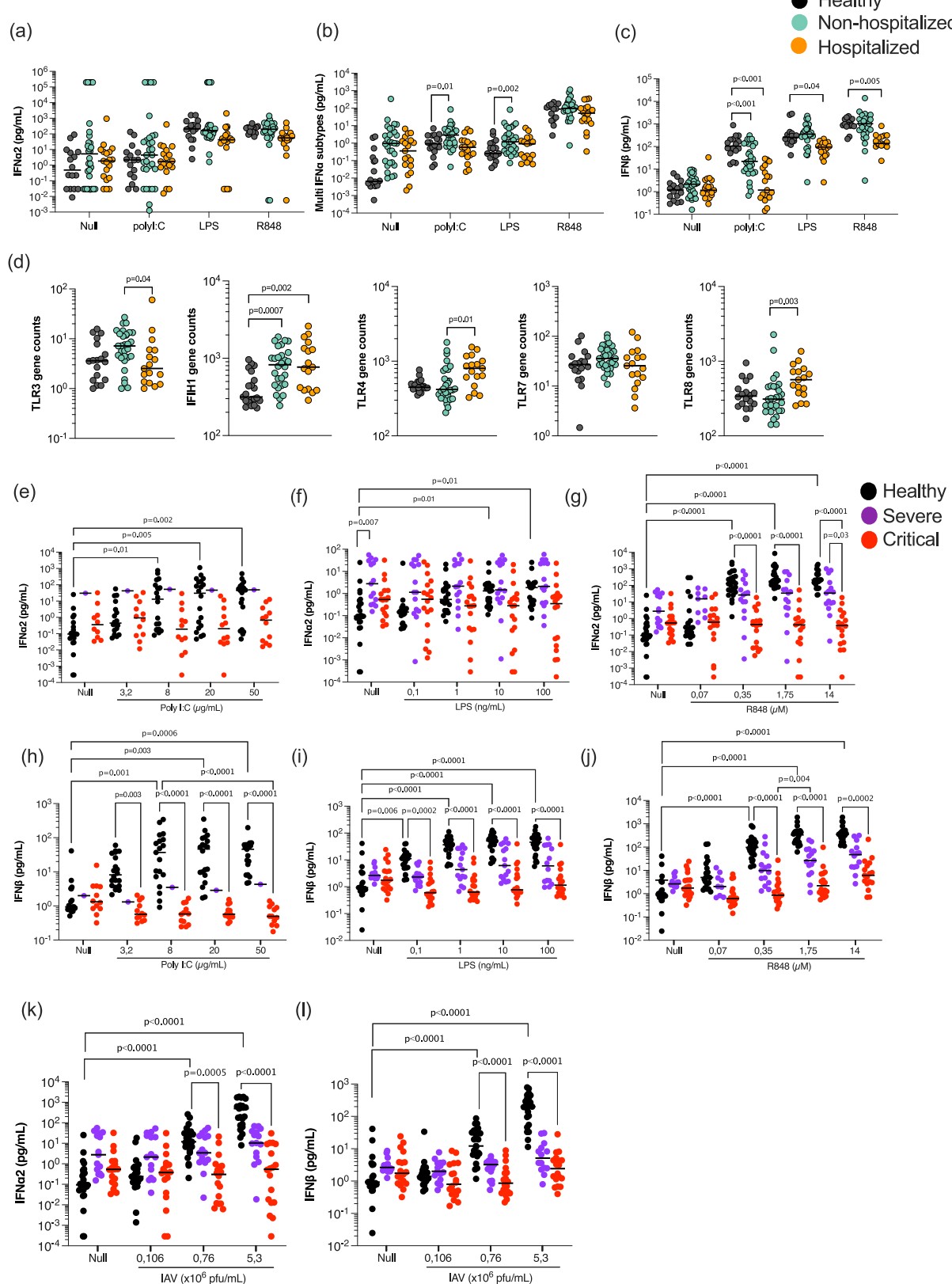

This highlights their already dysregulated state and is in line with the altered circulating IFN-I levels we observed. PolyI:C revealed the most distinct clusters, with hospitalized patients separating from controls along dimensions 1 and 2 (Fig. 5a).

To examine differences at a biological pathway level, we applied gene set enrichment analysis (GSEA) to each stimulation condition specifically comparing healthy controls *versus* non-hospitalized (Fig. 5b), control versus hospitalized (Fig. 5c), and non-hospitalized *versus* hospitalized (Fig. 5d). The most significant pathway differences were observed in the healthy *versus* moderate non-hospitalized comparison. As expected from our previous findings and in line with circulating IFN-I measurements, the null state of moderate patients

**Fig. 3 | Induction of IFN-I response is perturbed in hospitalized patients.**
**a** IFNα2, **b** multi-IFNα subtypes (equivalent IFNα17), and **c** IFNβ were measured by digital ELISA in healthy controls (n = 20), non-hospitalized (n = 30) and hospitalized (n = 18) COVID-19 patients after 22 h of whole blood stimulation with Poly:IC, LPS and R848. **d** mRNA of TLR3/4/7/8 and IFIH1 whole blood gene expression in healthy controls, non-hospitalized, and hospitalized COVID-19 patients. IFNα2 (**e–g**, **k**) and INFβ (**h–j**, **l**) responses in healthy donors (n = 24), severe (n = 11), and critical (n = 20) COVID-19 patients after variable dose stimulation with Poly:IC (**e**, **h**), LPS (**f**, **i**), R848 (**g**, **j**) and influenza virus (**k**, **l**). Black lines represent the medians. P values were determined either by multimodal regression models incorporating age and sex or by Kruskal–Wallis test followed by Dunn's post hoc test for multiple comparisons. Healthy controls = black, non-hospitalized COVID-19 patients = green and hospitalized COVID-19 patients = orange, severe = purple, and critical patients = red. N = number of individual patients included. Source data are provided as a Source Data file.

showed upregulation of type I and II IFN pathways, and these pathways were preferentially upregulated after R848 stimulation in controls (Fig. 5b, in black). Non-hospitalized COVID-19 patients also showed upregulated IL-17 responses, prostaglandin, and TGFB and HIF signaling to Poly:IC, LPS, and R848 stimulation (Fig. 5b, in green). Gene pathway alterations in hospitalized COVID-19 patients revealed consistently perturbed myeloid activation in unstimulated conditions and after Poly:IC stimulation, when compared with controls (Fig. 5c) and non-hospitalized patients (Fig. 5d). Additional pathways upregulated in hospitalized patients included coagulation, complement system, and TLR and MAPK signaling (Fig. 5c and d, in orange).

## IFN-I stimulation drives non-canonical inflammatory signaling in hospitalized patients only

Finally, after identifying perturbations in baseline and induced IFN responses, we wanted to assess the direct signaling response to IFN-I in COVID-19 patients. For this, we performed the same standardized whole blood ex vivo stimulation with recombinant IFNα2 and measured gene expression by Nanostring as described in the methods. Application of an ISG Z score showed that non-hospitalized and hospitalized patients were unable to induce a classical ISG response following direct IFNα stimulation, as was also the case with Poly:IC and R848 stimulations (Fig. 6a). This was likely explained by the already elevated ISG score in the Null condition of these patients. To apply a less biased analysis, we defined all genes that were significantly induced by IFNα stimulation across the entire cohort (q value < 0.05, Log 1.3-fold change compared to Null condition). Application of these 200 genes (Table S3) to a heat map clustered by gene response and grouped by clinical category revealed interesting differences (Fig. 6b). 50% of the genes were only induced in controls, and pathway analysis showed that these genes were classically involved in IFN-I and anti-viral responses (Fig. 6c), as exemplified by MX1 (Fig. 6d). An additional group of genes was upregulated in both controls and non-hospitalized patients, that included TLR3, TLR7, CXCL10, CXCL11, and HLA molecules (Fig. 6b, e). Most interestingly, the third cluster of genes was differentially expressed in hospitalized patients only, with pathway analysis showing this to consist largely of an inflammatory response (Fig. 6f, g). Many of these genes were downregulated after IFNα stimulation in controls and moderate patients, but not in hospitalized patients, as exemplified by IL1R1 (Fig. 6g). One hospitalized patient was identified to be positive for anti-IFN autoantibody (indicated by an * on the heat map, Fig. 6b). Strikingly this individual patient clearly lacked a classical ISG response to ex vivo IFNα stimulation, but their response to this non-canonical inflammatory activity was not affected.

## Discussion

Type I interferons mediate the major innate anti-viral immune activities through the activation of hundreds of genes. However, due to the complex regulation of their diverse functions, as well as their inhibition by many viruses, the precise role and impact of IFN-I in disease pathogenesis are not always evident. In COVID-19, the protective importance of IFN-I was evidenced by the identification of negative impacts on early IFN-I as strong risk factors for severe disease. These include neutralizing autoantibodies against IFNα, IFNω, and IFNβ[4,5], as well as inborn errors of immunity in IFN-mediated pathways including

TLR3[3] and TLR7[8]. Additional genetic evidence in support of a protective role for IFN-I in COVID-19 has also come from genome-wide association studies. These identified genome-wide associations between severe COVID-19 and gene clusters in IFNAR2 (subunit of IFN receptor), near the gene encoding tyrosine kinase 2 (TYK2) (associates with plasma domain of IFNAR), and in a gene cluster encoding antiviral restriction enzyme activators (OAS1, OAS2, OAS3) (induced by IFN-I)[27]. Highlighting the complex role of ISGs in viral responses, and the need for careful interpretation, mendelian randomization showed a link between life-threatening disease and low expression of IFNAR2, but high expression of TYK2[27]. However, although these associations had genome-wide significance, the effect sizes were relatively modest (OR 1.3–1.6). Despite these collective studies highlighting how critical IFN-I immunity is in dictating COVID-19 outcomes, cumulatively, they cannot account for the majority of the defective IFN-I responses observed in severe COVID-19. Furthermore elevated IFNβ has been recently implicated in long COVID-19, emphasizing the need to better understand the regulation of IFN-I during infection with SARS-CoV-2.

Our study adds further knowledge to why IFN-I responses are defective in severe COVID-19 through two major findings: first, we show that a combination of decreased circulating pDC numbers and dysregulated monocytes in critical disease is a major cause of ineffective IFNα secretion and second, IFN-I stimulation of leukocytes from severe COVID-19 patients promotes an inflammatory response that was not observed in moderate patients.

Undoubtedly some of the early confusion around the role of systemic IFN in COVID was due to the different assays used to measure the cytokine directly. A meta-analysis comparing 15 studies that used either ELISA, single molecular array (Simoa) digital ELISA, Luminex, electrochemiluminescent, flow cytometry bead-based immunoassay, and microfluid immunoassay fluorescence detection techniques, showed no significant differences in plasma levels between different disease severities[23]. However, our results presented here highlight the importance of using sufficiently sensitive assays for measuring IFNα proteins, as physiological concentrations are often below pg/mL levels[24,25]. When such assays are applied to COVID-19 patient samples, more consistent results were observed, with severe patients overall showing lower levels of circulating IFNα[2,15,28–31]. Nevertheless, our results in more heterogenous patient populations with co-morbidities and older ages revealed a larger range of IFNα levels, highlighting the challenges of comparing across studies and translating to possible clinical applications. The presence of 13 IFNα subtypes can also lead to confusion, with many assays used not reporting the subtype measured. A recent in vitro study reported different anti-SARS-CoV-2 functional activity between the diverse IFNα subtypes, and here we observed differences in IFNα2 protein plasma levels, but not total subtype levels, between moderate and severe disease. New experimental tools will be needed to fully understand the different roles of all IFNα subtypes in COVID-19 patients, which given their previously reported evolutionary selection[32], may be relevant for other viral infections.

Poor results in randomized placebo-controlled treatment studies have further divided opinions on the importance of IFN-I in COVID-19[9,10]. No clinical benefit was observed in these studies where either IFNα or IFNβ were given alone, or in combination with anti-virals. However, a major caveat of these studies is the late initiation of

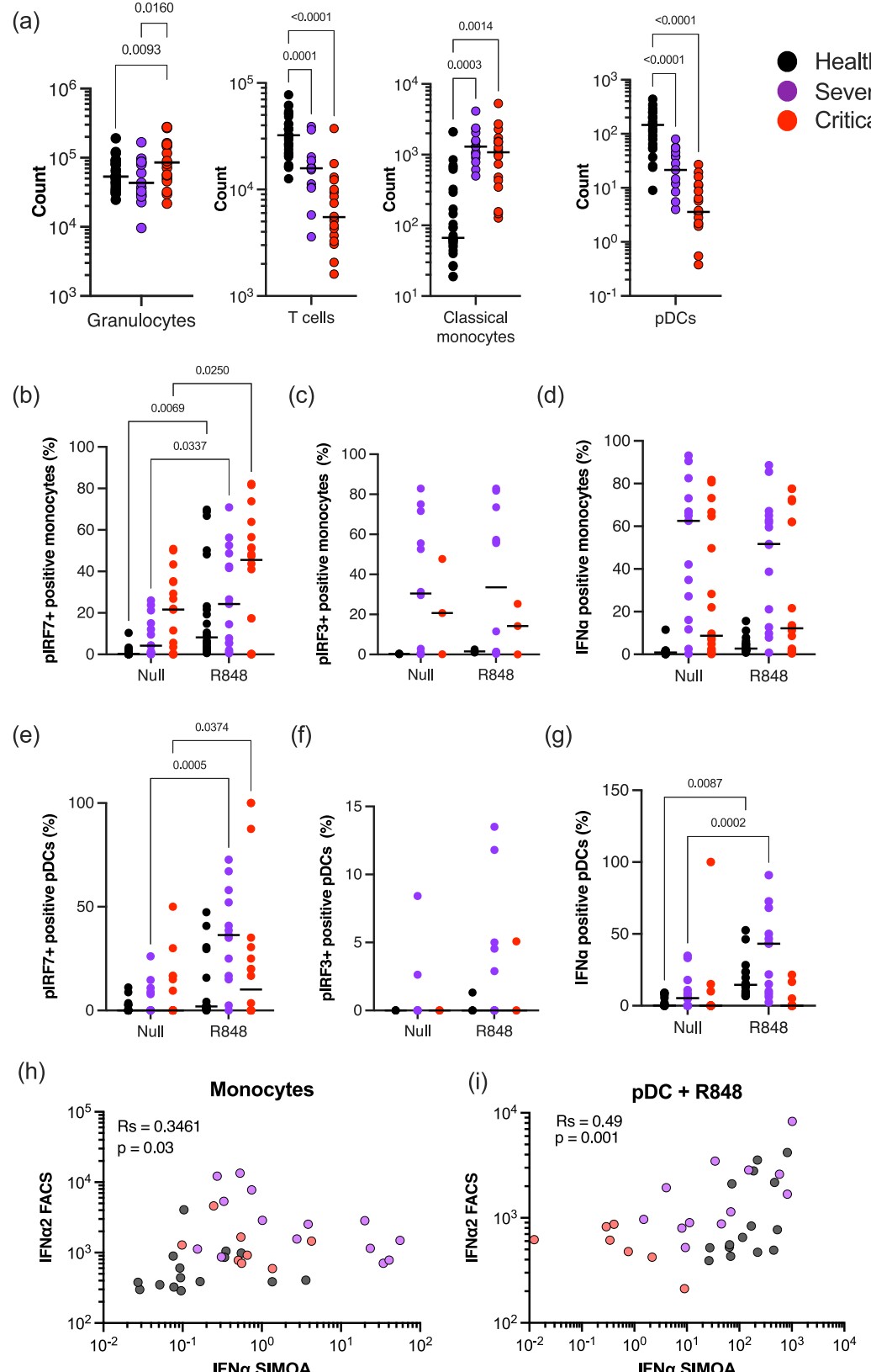

**Fig. 4 | Perturbed intracellular IFN-I responses in COVID-19 patients. a** Counts of granulocytes, T cells, classical monocytes, and pDCs in the blood of healthy controls ($n = 29$), severe ($n = 11$), and critical ($n = 20$) COVID-19 patients. Percentages of **b**, **e** phosphorylated IRF7, **c**, **f** phosphorylated IRF3, or **d**, **g** intracellular IFNα positive monocytes (**b**–**d**) and pDCs (**e**–**g**) in the blood of healthy controls, severe and critical COVID-19 patients in the absence of stimulation or after overnight stimulation with R848. Correlation of intracellular IFNα2 measured by flow cytometry with plasma IFNα2 levels measured by Simoa in **h** monocytes without stimulation or in **i** pDCs after R848 stimulation. *P* values were determined by Kruskal–Wallis test followed by Dunn's post hoc test for multiple comparisons. *N* = number of individual patients included. Source data are provided as a Source Data file.

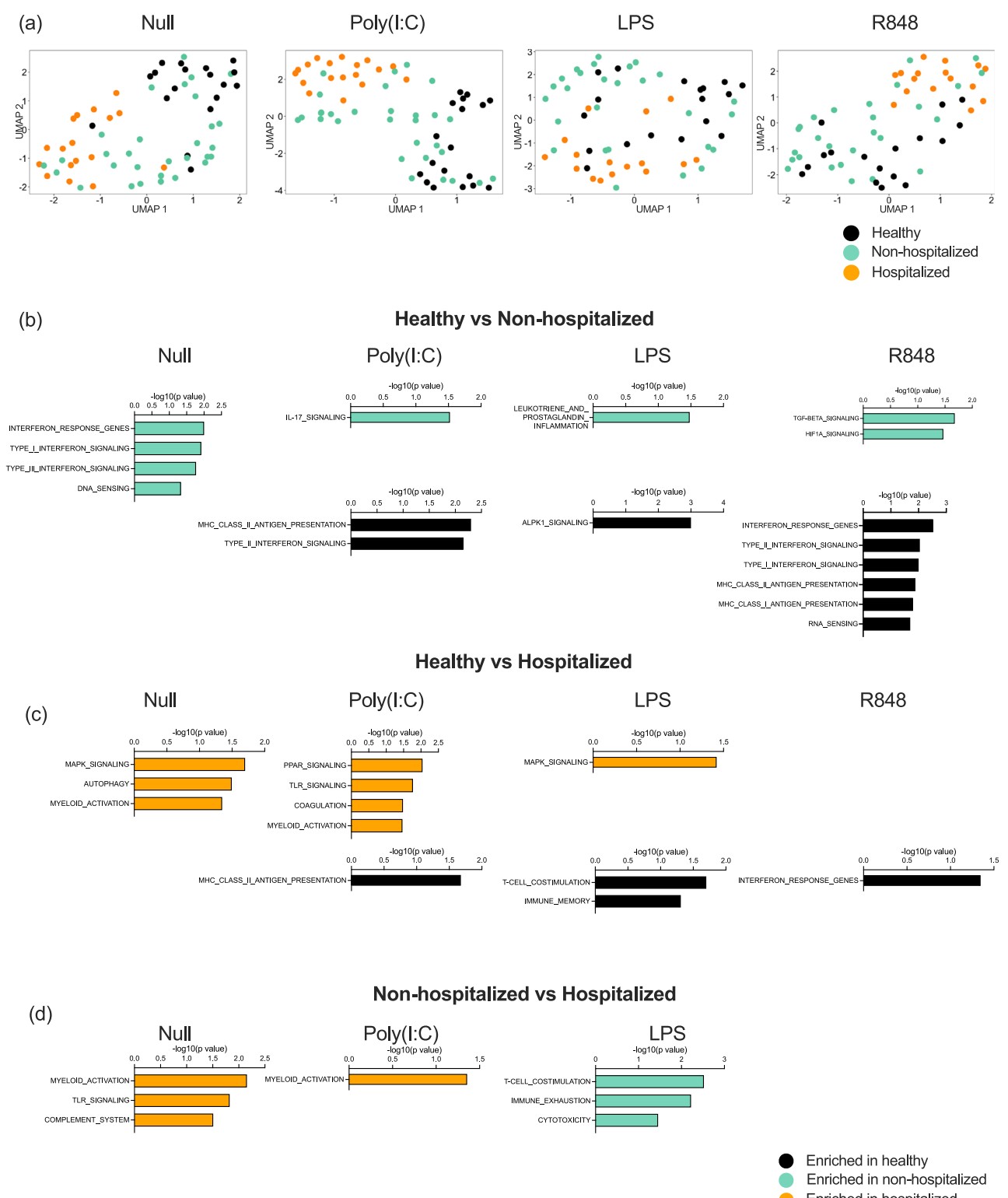

**Fig. 5 | Induced gene expression differences in moderate and hospitalized Covid-19 patients. a** UMAP plots of Nanostring gene expression in the Null control, and after whole blood stimulation with Poly:IC, LPS, and R848. Gene expression pathways enriched in healthy donors (black, $n = 16$), moderate non-hospitalized COVID-19 patients (green, $n = 30$), or hospitalized COVID-19 (orange, $n = 18$) patients after comparisons between **b** healthy and moderate, **c** healthy and hospitalized, **d** moderate and hospitalized, after whole blood stimulation with Poly:IC, LPS, and R848. $N$ = number of individual patients included. Source data are provided as a Source Data file.

treatment when patients were unlikely to benefit from further antiviral signaling. Moreover, at this stage of infection, IFN-I may even suppress adaptive immunity, further compromising any apparent clinical benefit. This is supported by results from our study where

COVID-19 patients with elevated ISG expression did not respond to ex vivo IFN stimulation, and hospitalized patients in particular showed an inflammatory gene expression pattern that was not downregulated by IFN-I. Direct testing of this hypothesis may be

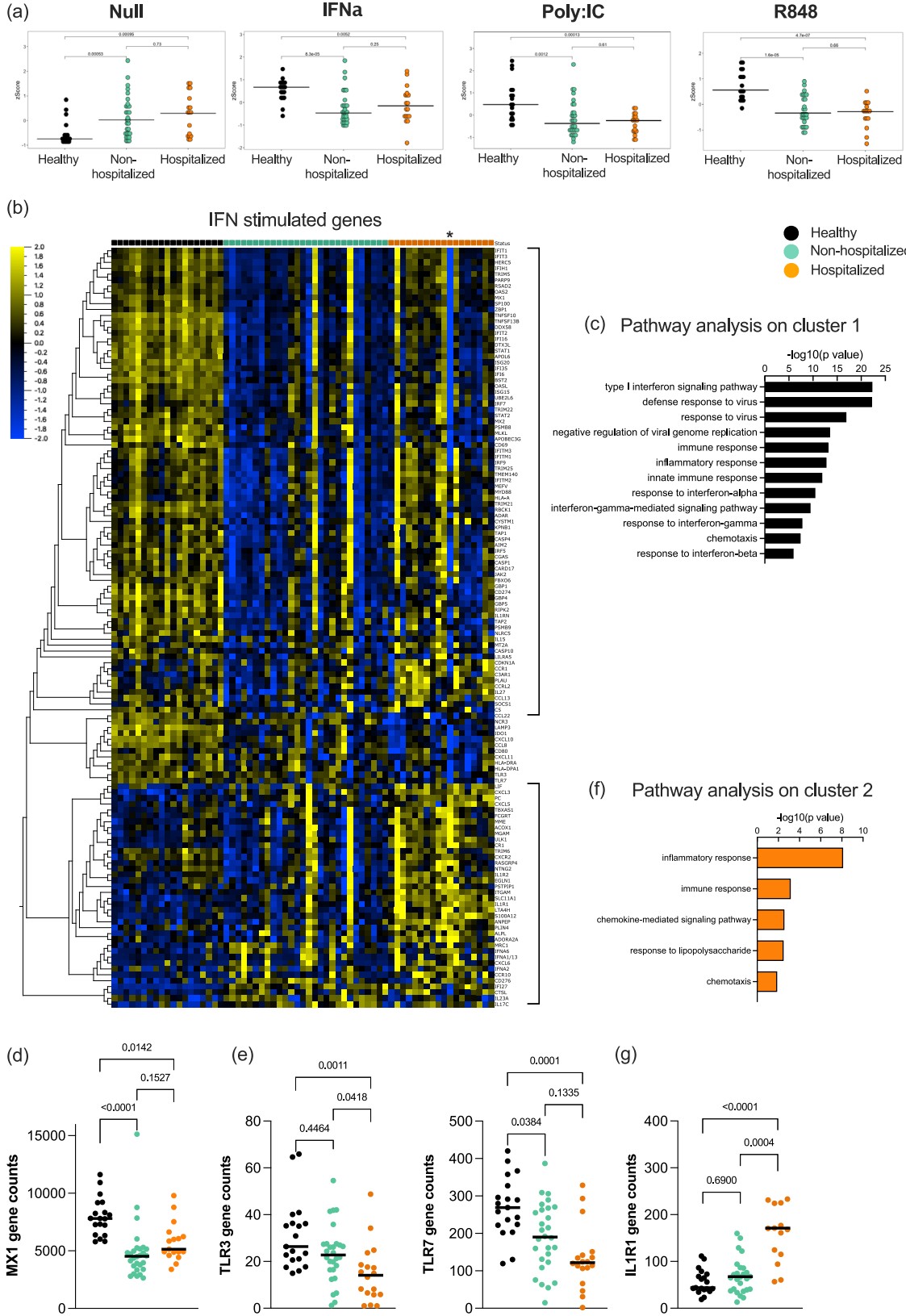

(a) Null, IFNa, Poly:IC, R848

(b) IFN stimulated genes

(c) Pathway analysis on cluster 1

(f) Pathway analysis on cluster 2

(d) MX1 gene counts (e) TLR3 gene counts, TLR7 gene counts (g) IL1R1 gene counts

provided by clinical studies testing early intervention prior to the appearance of symptoms[33] or targeted to patients with known risk factors[34], or retrospective testing of pre-treatment IFN-I levels. Reflecting multiple perturbations in the IFN-I response with increasing COVID-19 severity was our striking observation that severe patients failed to secrete IFN-I proteins after stimulation with diverse viral agonists. This supports previous results showing a reduced ability of pDCs and monocytes from COVID-19 patients to respond to TLR stimulation[31]. IFNβ secretion was even more strikingly perturbed than IFNα, which was surprising given its low levels in patient plasma. However, the relatively low number of studies implicating this IFN subtype in COVID-19 are indicative of the challenges in accurately

**Fig. 6 | Downstream signaling response to IFNα is perturbed in hospitalized patients. a** Type I interferon gene signature (ISG) score calculated from Nanostring data in healthy controls (black, $n = 19$), moderate COVID-19 patients (green, $n = 28$), and hospitalized COVID-19 patients (orange, $n = 18$) in the Null control, and after stimulation with IFNα, Poly:IC, and R848. **b** Heat map of all IFN stimulated genes identified from Nanostring data in healthy controls (black), moderate COVID-19 patients (green), and hospitalized COVID-19 patients (orange). Pathways identified to be enriched after IFNα stimulation in **c** healthy donors and **f** hospitalized COVID-19 patients. Nanostring gene expression of **d** MX1, **e** TLR3 and TLR7, and **g** IL1R1 in whole blood of healthy controls (black), moderate COVID-19 patients (green), and hospitalized COVID-19 patients (orange). Black lines represent the medians. $P$ values were determined with the Kruskal–Wallis test followed by Dunn's post hoc test for multiple comparisons. $N$ = number of individual patients included. Source data are provided as a Source Data file.

detecting and quantifying this interferon in the blood, and may reflect its greater importance in the infected tissues.

Why such broad IFN protein responses are blunted in COVID-19 patients remains unexplained, beyond the impact of neutralizing autoantibodies[4,5] and inborn errors of IFN-I immunity[3,8]. It is possible that lower concentrations of autoantibodies, below detection limits of current assays, may be clinically relevant in certain patients but our intra-cellular cytokine data suggests this is not the case. While the SARS-CoV-2 virus has been shown to interfere with many aspects of host immunity[35], we studied blood immune cell responses where the presence of the virus was undetectable by droplet digital PCR. Therefore, it is unlikely that this phenotype could be directly attributed to viral interference. While pDCs numbers were reduced with increasing severity, they still remained capable of producing IFNα as measured intracellularly, in line with previous in vitro studies[36]. In contrast, monocytes that were increased in the circulation with severity appeared to express IFNα intracellularly in the steady state but did not increase production or secretion after TLR stimulation. Further functional analysis of intracellular pathways in isolated cells from severe COVID-19 patients will be required to fully understand this phenotype, which may provide targets for new treatment strategies.

## Methods

### Patient cohorts
Clinical cohorts are summarized in Table S1. Healthy controls ($n = 14$) and patients acutely infected with the SARS-CoV-2 virus ($n = 144$) were previously described[2] (Table S1, Supplemental Dataset 2, 3), or recruited as inpatients or as outpatients following receipt of a positive SARS-CoV-2 nasopharyngeal swab PCR test at St. James's Hospital (SJH) in Dublin, Ireland ($n = 136$) (Table S1, Supplemental Dataset 3, 4) from March to June 2020. Ethical approval was obtained for the study from the Tallaght University Hospital (TUH)/SJH Joint Research Ethics Committee (reference REC 2020-03). Severity grades were based on admission and supplemental oxygen requirements at the time of sampling. Moderate patients did not require hospitalization at any timepoint. Hospitalized patients requiring supplemental oxygen via nasal cannula (maximal supplemental oxygen flow of up to 6 L/min) were considered severe, with critical disease classified as requiring more than 6 L of oxygen per minute, either delivered via high-flow nasal oxygen (Airvo) or a venturi mask, a clinical definition previously defined[3,4]. Additional hospitalized patients (severe and critical cases) were also recruited for cellular and functional assays (Table S1, Supplemental Dataset 5) from Hopital Cochin and Hopital Bichat, Paris under clinical study protocols in the setting of the local RADIPEM biological samples collection, derived from samples collected in routine care as previously described[2], or from the INSERM-sponsored French COVID-19 clinical study (NCT04262921). Biological collection and informed consent were approved by the Direction de la Recherche Clinique et Innovation and the French Ministry of Research (nos. 2019-3677, 2020-A00256-33). The studies conformed to the principles outlined in the Declaration of Helsinki, and received approval by the appropriate Institutional Review Boards (Cochin-Port Royal Hospital, Paris; no. AAA-2020−08018 and Comité de protection des personnes Ile de France VI; no. 2020-A00256-33). Plasma samples were obtained from COVID-19 patients ($n = 311$) for cytokine analysis and for autoantibody analysis ($n = 146$), and whole blood for immune stimulations

($n = 79$) and cellular phenotyping ($n = 31$) from subgroups. Written informed consent was obtained from all study participants. Healthy controls ($n = 63$) were asymptomatic adults, matched with individuals with COVID-19 on age (±5 years), who had a negative SARS-CoV-2 RT–PCR test at the time of inclusion.

### Cytokine assays
Prior to protein analysis, plasma and TruCulture supernatants were treated in a P3 laboratory for viral decontamination using a protocol previously described for SARS-CoV[37] which we validated for SARS-CoV-2. Briefly, samples were treated with TRITON X100 (TX100) 1% (v/v) for 2 h at RT. IFNα2 and IFNβ proteins were quantified by Simoa assays developed with Quanterix Homebrew kits as previously described[2]. Multiple IFNα protein subtypes (named Multi-IFNα subtypes) were measured with an IFNα multi-subtype prototype assay (Quanterix), using IFNα17 (PBL Assay Science) as a reference recombinant standard. For the IFNα2 assay, the BMS216C (eBioscience) antibody clone was used as a capture antibody after coating on paramagnetic beads (0.3 mg/mL), and the BMS216BK already biotinylated antibody clone was used as the detector at a concentration of 0.3 µg/mL. The SBG revelation enzyme concentration was 150 pM. Recombinant IFNα2c (eBioscience) was used as a calibrator. The SBG revelation enzyme concentration was 150 pM. The recombinant protein (PBL Assay Science) was used to quantify IFNγ concentrations. For the IFNβ assay, the 710322-9 IgG1, kappa, mouse monoclonal antibody (PBL Assay Science) was used as a capture antibody after coating on paramagnetic beads (0.3 mg/mL), and the 710323-9 IgG1, kappa, mouse monoclonal antibody (PBL Assay Science) was biotinylated (biotin/antibody ratio = 40/1) and used as the detector antibody at a concentration of 1 µg/mL. The SBG revelation enzyme concentration was 150 pM. The recombinant protein (PBL Assay Science) was used to quantify IFNβ concentrations. The limit of detection (LOD) of these assays were 0.6–2 fg/mL for IFNα2, 0.6 pg/mL for IFNβ, 0.6 fg/mL for the IFNα multi-subtype. An additional 44 cytokines and chemokines, including IFNα2 were measured in plasma and TruCulture supernatants with a commercial Luminex multi-analyte assay (Biotechne, R&D systems).

### Functional immune assays
For whole blood stimulation, TruCulture tubes (RBM) containing Poly:IC (20 µg/mL), R848 (1 µM), LPS-EB (ultrapure) (10 ng/mL) (all Invivogen), and IFN-α2 (Intron A, Merck) dissolved in 2 mL of buffered media were batch produced as previously described[38]. Tubes were thawed at room temperature and 1 mL of fresh blood was distributed into each tube within 15 min of collection. Tubes were mixed by inverting them and incubated at 37 °C for 22 h in a dry block incubator. After the incubation time, a valve was manually inserted into the tube to separate the supernatant from the cells. The supernatant was collected, aliquoted, and immediately stored at −80 °C for protein secretion analysis. Cell pellets of the TruCulture tubes were resuspended in 2 mL of Trizol LS (Sigma) and tubes were vortexed for 2 min at 2000 rpm and stored at −80 °C for gene expression analysis.

### Flow cytometry
Whole blood was retrieved and incubated in PBS containing 2% fetal calf serum and 2 mM EDTA (FACS buffer) for 10 min at 37 °C. After centrifugation, the supernatant was removed, and 1× RBC lysis buffer

(Biolegend) was added for 15 min at room temperature. Cells were washed in PBS and then incubated with a viability stain (Zombie-Aqua, BioLegend) for 10 min at 4 °C. After washing, the cells were resuspended in FACS buffer and stained with an extracellular mix containing the antibodies shown in Table S2. For intracellular staining, Fixation/ Permeabilization Solution Kit (BD Cytofix/Cytoperm) was used according to the manufacturer's protocol. Briefly, the cells were fixed for 10 min at 4 °C with 100 μL of the Fixation/Permeabilization solution and then washed and stained in 100 μL of the BD Perm/Wash Buffer containing the intracellular mix of antibodies for 1 h at 4 °C. Data acquisition was performed on a FACS LSR flow cytometer using FACSDiva software (BD Biosciences, San Jose, CA). FlowJo software (Treestar, Ashland, OR) was used to analyze data.

## Nanostring gene expression arrays

Total RNA was extracted from Trizol-stabilized cell pellets using NucleoSpin 96 miRNA kit (Macherey-Nagel). RNA concentrations were measured using Quantifluor RNA system kit (Promega) and RNA integrity numbers were determined using the Agilent RNA 6000 Nano kit (Agilent Technologies). Total RNA samples were analyzed using the Human Host Response panel profiling 800 immunology and host response-related human genes (Nanostring). Gene expression data were normalized as previously described using nCounter software (Nanostring)[39].

## Autoantibody measurement

Recombinant human (rh) IFNα2 (Miltenyi Biotec, reference number 130-108-984) or rhIFNω (Merck, reference number SRP3061), was first biotinylated with EZ-Link Sulfo-NHS-LC-Biotin (Thermo Fisher Scientific, catalog number A39257), according to the manufacturer's instructions, with a biotin-to-protein molar ratio of 1:12. The detection reagent contained a secondary antibody [Alexa Fluor 647 goat anti-human IgG (Thermo Fisher Scientific, reference number A21445)] diluted in Rexxip F (Gyros Protein Technologies, reference number P0004825; 1:500 dilution of the 2 mg/mL stock to yield a final concentration of 4 μg/mL). Buffer phosphate-buffered saline, 0.01% Tween 20 (PBS-T), and Gyros Wash buffer (Gyros Protein Technologies, reference number P0020087) were prepared according to the manufacturer's instructions. Plasma or serum samples were then diluted 1:100 in PBS-T and tested with the Bioaffy 1000 CD (Gyros Protein Technologies, reference number P0004253) and the Gyrolab xPand (Gyros Protein Technologies, reference number P0020520).

## ISG scores

Two separate ISG scores were calculated due to different available data sets. The first was previously described[2]. Briefly quantitative reverse transcription polymerase chain reaction (qPCR) analysis was performed using the TaqMan Universal PCR Master Mix (Applied Biosystems) using cDNA derived from 40 ng total RNA extracted from whole blood cells. Using TaqMan probes for *IFI27* (Hs01086370_m1), *IFI44L* (Hs00199115_m1), *IFIT1* (Hs00356631_g1), *ISG15* (Hs00192713_m1), *RSAD2* (Hs01057264_m1), and *SIGLEC1* (Hs00988063_m1), the relative abundance of each target transcript was normalized to the expression level of *GAPDH* (Hs03929097_g1). qPCR was performed in duplicate using the LightCycler VIIA7 System (Roche). The RQ value was equal to 2ΔΔct where ΔΔct is calculated by (CT target-CT *GAPDH* test sample-(CT target-CT *GAPDH* calibrator sample. ISG score was considered as the mean of the six selected genes.

An ISG score was also calculated from the Nanostring Null data set based on a previously defined IFN-I gene signature of whole blood stimulated with recombinant IFNβ[40]. The genes included; *BST2, CCL8, CCR1, CXCL10, IFI35, IFIH1, IFITM1, IRF7, MX1, STAT2,* TNFSF10, and *TNFSF13B*, and the ISG score was calculated as the average gene level *Z* scores per sample using log2-fold change.

## Type I IFN cytopathic assay

Type I IFN activity was measured as previously described[2] by determining the cytopathic reduction (i.e., protection of Madin–Darby bovine kidney cells against cell death after infection with vesicular stomatitis virus) afforded by patient serum. A reference of human IFNα, standardized against the National Institutes of Health reference Ga 023–902-530, was included with each titration. IFNα activity in normal healthy serum is IU/mL.

## Statistical analysis

GraphPad Prism (Version 9) and R were used for statistical analysis. We applied multinomial logistic regression models between patient groups using IFN response phenotypes and accounting for impacts of age and sex (known factors associated with COVID-19 severity) using the "nnet" (version 7.3) R package. For comparison of patient subgroups with smaller sample sizes, two-sided non-parametric Kruskal–Wallis tests, followed by Dunn's post-test for multiple group comparisons were performed. Correlations between the different assays were calculated using the Spearman test. UMAP plots were performed with "M3C" R package (v1.10.0). Gene set enrichment analysis (GSEA) was performed as previously described[2] using a pathway data set built from the Nanostring Host response panel annotation file. Heatmaps were produced with Qlucore (version 3.5). Dot plots and correlation graphs were produced with GraphPad Prism (version 9).

## Reporting summary

Further information on research design is available in the Nature Portfolio Reporting Summary linked to this article.

## Data availability

All immune phenotype data is provided in the supplemental data and source data files. Source data are provided with this paper.

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

## Acknowledgements
This study was supported by the "URGENCE COVID-19" fundraising campaign of the Institut Pasteur (CoVarImm and Steroid Response), from the Agence Nationale de la Recherche (ANR-flash COVID-19), by the Laboratoire d'Excellence 'Milieu Intérieur' (grant no. ANR-10-LABX-69-01) (D.D.), the Fonds IMMUNOV for Innovation in Immunopathology (B.T.), and Science Foundation Ireland (C.O.F., C.N.C., and N.B.). We thank the STTAR-Bioresource of TCD-SJH-TUH COVID-19 bioresource which supported the collection of patient samples. N.S. is a recipient of the Pasteur–Roux–Cantarini Fellowship. C.O.F., N.C., and C.N.C. are part-funded by a Science Foundation Ireland (SFI) grant, Grant Code 20/SPP/3685. L.T. is supported by the Irish Clinical Academic Training (ICAT) Program, supported by the Wellcome Trust and the Health Research Board (Grant Number 203930/B/16/Z), the Health Service Executive, National Doctors Training and Planning and the Health and Social Care, Research and Development Division, Northern Ireland. N.B. is funded under the Science Foundation Ireland Phase 2 COVID-19 Rapid Response Call (20/COV/8487) and the Health Research Board COVID-19 Rapid Response Call (COV19e2020e053). The Laboratory of Human Genetics of Infectious Diseases is supported by the Howard Hughes Medical Institute, the Rockefeller University, the St. Giles Foundation, the National Institutes of Health (NIH) (R01AI088364 and R01AI163029), the National Center for Advancing Translational Sciences (NCATS), NIH Clinical and Translational Science Award (CTSA) program (UL1 TR001866), a Fast Grant from Emergent Ventures, Mercatus Center at George Mason University, the Fisher Center for Alzheimer's Research Foundation, the Meyer Foundation, the JPB Foundation, the French National Research Agency (ANR) under the "Investments for the Future" program (ANR-10-IAHU-01), the Integrative Biology of Emerging Infectious Diseases Laboratory of Excellence (ANR-10-LABX-62-IBEID), the French Foundation for Medical Research (FRM) (EQU201903007798), the ANRS-COV05, ANR GENVIR (ANR-20-CE93-003), ANR AABIFNCOV (ANR-20-CO11-0001) and ANR GenMISC (ANR-21-COVR-0039) projects, the European Union's Horizon 2020 research and innovation program under grant agreement No. 824110 (EASI-genomics), the Square Foundation, Grandir—Fonds de solidarité pour l'enfance, the Fondation du Souffle, the SCOR Corporate Foundation for Science, The French Ministry of Higher Education, Research, and Innovation (MESRI-COVID-19),

Institut National de la Santé et de la Recherche Médicale (INSERM), REACTing-INSERM and the University of Paris (J.L.C.). P.B. was supported by the MD–Ph.D. program of the Imagine Institute (with the support of the Fondation Bettencourt-Schueller). We thank the UTechS CB of the Center for Translational Research, Institut Pasteur for supporting Luminex and Nanostring analysis. We thank Quanterix for the provision of IFNα multi-subtype prototype assays. We acknowledge all healthcare workers involved in the diagnosis and treatment of patients in Hopital Cochin, Hopital Bichat, and St James's Hospital Dublin.

## Author contributions

N.S., C.P., V.B. generated the main data sets, analyzed and interpreted them. J.S., L.T., O.S., M.C.G., J.G., T.A.S., B.J., N.C. collected patient samples, curated and analyzed clinical data. B.C., T.D., A.R.P., N.Y., P.B., J.L.C. generated and analyzed results. V.R., V.S.A., D.D. performed analytical models, analyzed and interpreted data. C.O.F., C.N.C., N.M.B. and D.D. designed the overall study. D.D. wrote the main paper and coordinated the overall study. All authors discussed the results and implications and commented and edited on the manuscript.

## Competing interests

J.L.C. is an inventor on patent application PCT/US2021/042741, filed 22 July 2021, submitted by The Rockefeller University, which covers diagnosis of, susceptibility to, and treatment of viral disease and viral vaccines, including COVID-19 and vaccine-associated diseases. O.S. is an inventor on provisional patent no. US 63/020,063 entitled "S-Flow: a FACS-based assay for serological analysis of SARS-CoV2 infection" submitted by Institut Pasteur. The other authors declare no competing interests.

## Inclusion and Ethics statement

Researchers involved in identification, recruitment, and sampling of patients in the clinical centers in Ireland and France have been included as co-authors.

## Additional information

**Nikaïa Smith** [1,21], **Céline Possémé** [1,21], **Vincent Bondet** [1,21], **Jamie Sugrue** [2], **Liam Townsend** [3,4], **Bruno Charbit** [5], **Vincent Rouilly** [6], **Violaine Saint-André** [1,7], **Tom Dott** [5], **Andre Rodriguez Pozo** [1], **Nader Yatim** [1], **Olivier Schwartz** [8], **Minerva Cervantes-Gonzalez** [9,10,11], **Jade Ghosn** [9,11], **Paul Bastard** [12,13,14,15], **Jean Laurent Casanova** [12,13,14,15,16], **Tali-Anne Szwebel** [17], **Benjamin Terrier** [17], **Niall Conlon** [18,19], **Cliona O'Farrelly** [2,19], **Clíona Ní Cheallaigh** [3,4], **Nollaig M. Bourke** [20] & **Darragh Duffy** [1,5] ✉

[1]Translational Immunology Unit, Institut Pasteur, Université Paris Cité, Paris, France. [2]School of Biochemistry and Medicine, Trinity Biomedical Sciences Institute, Trinity College Dublin, Dublin, Ireland. [3]Discipline of Clinical Medicine, School of Medicine, Trinity Translational Medicine Institute, Trinity College Dublin, Dublin, Ireland. [4]Department of Infectious Diseases, St James's Hospital, Dublin, Ireland. [5]Institut Pasteur, Université Paris Cité, CBUTechS, Center for Translational Research, Paris, France. [6]Datactix, Bordeaux, France. [7]Department of Computational Biology, Université Paris Cité, Bioinformatics and Biostatistics HUB, Institut Pasteur, Paris, France. [8]Department of Virology, Virus and Immunity Unit, Institut Pasteur, Paris, France. [9]Infectious and Tropical Diseases Department, AP-HP, Hôpital Bichat, Paris, France. [10]Epidemiology, Biostatistics and Clinical Research Department, AP-HP, Hôpital Bichat, Paris, France. [11]Université de Paris, INSERM, IAME, UMR 1137 Paris, France. [12]Department of Pediatrics, Necker Hospital for Sick Children, AP-HP, Paris, France. [13]Laboratory of Human Genetics of Infectious Diseases, Necker Branch, INSERM U1163, Necker Hospital for Sick Children, Paris, France. [14]Université Paris Cité, Imagine Institute, Paris, France. [15]St. Giles Laboratory of Human Genetics of Infectious Diseases, Rockefeller Branch, The Rockefeller University, New York, NY, USA. [16]Howard Hughes Medical Institute, New York, NY, USA. [17]Institut Cochin, APHP, Paris, France. [18]Department of Immunology, St James's Hospital, Dublin, Ireland. [19]Discipline of Immunology, School of Medicine, Trinity College Dublin, Dublin, Ireland. [20]Discipline of Medical Gerontology, School of Medicine, Trinity Translational Medicine Institute, Trinity College Dublin, Dublin, Ireland. [21]These authors contributed equally: Nikaïa Smith, Céline Possémé, Vincent Bondet. ✉e-mail: darragh.duffy@pasteur.fr

