## [Peer Review File · Nature Communications]

Detection of defective type I interferon immunity is associated with increasing COVID-19 severityREVIEWER COMMENTS

Reviewer #1 (Remarks to the Author):

In this manuscript, Smith, Poss  m   and Bondet et al studied type I interferon responses in samples from several COVID-19 patient cohorts in depth. First, the authors confirmed their previous findings (PMID: 32661059) showing that IFN responses were impaired in critical patients, where the concentration of IFN-alpha found in plasma samples inversely correlated with disease severity. For this, they highlighted the need to use digital ELISA, a highly sensitive assay to measure interferon concentration, as Luminex couldn't pick differences between patient groups. Moreover, and contrary to the Luminex data, data from digital ELISA correlated well with the "ISG score" (based on the relative quantification of 6 prototype ISG expression) and interferon activity (as measured by protection against virus-induced cytopathic effects). They further showed that blood cells from critical patients globally responded less well to various stimuli (poly(I:C), LPS, R848) and confirmed that their pDC count was significantly lower, whereas monocyte counts were increased. Using the nanostring technology on 800 immunology and host response related genes on patient cells exposed or not to IFN-alpha, poly(I:C) and R848, they showed a perturbation at baseline of immune responses and a lack of ISG response following stimulation in COVID-19 patients. Moreover, they could detect increased inflammatory responses in hospitalized patients.

This is a highly interesting area of research and the manuscript presents an impressive amount of data. However, in order to improve the manuscript and facilitate its reading, some clarifications in the text are required. Moreover, some of the data sets used to generate some of the main figures should be shown in the supplemental figures. Please find these recommendations below.

For clarity, in the Results section and in the figure legends, when mentioning the cohorts, the authors should always refer to the corresponding supplementary tables (this was done only for Table S3 and S5).

The authors took into account the sex and age of patients in their statistical analyses. However, the sampling time (number of days post symptoms), when available, did not seem to have been taken into account in the analyses for Cohort 1 (Fig. 1, S1). Importantly, for this cohort (Table S2) the average of days post symptoms were different between the 3 categories (moderate, severe, critical) (see review figure - Days post symptom Cohort 1 and 3), and although the differences appeared not significant (using a Mann-Whitney test), it might somehow impact the findings. The authors should acknowledge this. Could the authors plot the IFN concentrations according to the number of days post symptoms for each sample (and using the disease severity color code)? This might help discard the possibility that the time of sampling parameter impacted the data.

In the third cohort (Table S4), according to date of sample and date of onset of symptoms, some samples were harvested >60 days post-symptom, in particular for patients with moderate disease (see review figure - Days post symptom Cohort 1 and 3). One might wonder why the authors kept such outliers for moderate disease and whether they impacted the results.

The discrepancy between Luminex and digital ELISA data is worrying and, as such, an important finding as it could explain some of the discrepancies found in the literature. In Fig 1c (Luminex analysis) compared to 1a – b (digital ELISA) (data obtained with Cohort 1), the

concentrations measured were completely different; between 10^0 to 10^3 pg/mL for instance in the healthy group with Luminex and between 10^{-1} and 10^{-2} pg/mL with digital ELISA. How could this be explained? As the values are quite high with the Luminex analyses, a problem with detection limit is not what first comes to mind. What was the limit of detection for the Luminex assay?

The experiments performed in Fig. 1 need to be better explained and the data provided. Notably, the ISG score and functional cytopathic experiments were not explained. For this, an article (PMID: 32661059) was quoted, but one has to download the supplemental methods from that paper to get the information. The authors should define in detail how the ISG score was obtained and provide the actual data (in a supplemental figure) for the expression measure of the 6 chosen ISGs (data corresponding to Fig. 1d-f). Moreover, it was unclear what the calibrator sample (used for 2-DDCt calculation) was. The authors should also define how the functional cytopathic effect experiment was performed (for this, reference 2, PMID: 32661059, actually referred to a paper from 1979 without further details) and the data should be provided (i.e. data corresponding to Fig. 1g-i).

The ISG score measured using the nanostring technology (Fig. 2b-c) should also be explained, and the data provided. Moreover, could the authors define the paired blood samples?

Fig. 2c-d, S2f-g: there are no units on the y-axis and y-axis, respectively

In the following paragraph, the figures are mislabeled, S2b should be replaced by S2e "... This showed low transcriptional levels of all measured IFNA subtypes as expected (Fig. S2b). Despite these low baseline levels, among the 7 IFNA subtypes examined we did observe some subtype differences, with notably higher levels of IFNA6 in both COVID-19 patient groups and IFNA1/13 and IFNA5 only in the hospitalized group (Fig. S2b). IFNA2 was notably no different between all three groups (Fig. S2e)."

Fig 2c-d description in the main text: (>95 considered positive): please specify the units

Please correct the figure numbers in the following paragraph: "Additional correlation analysis between cytometry measured intracellular IFN α , and digital ELISA measured plasma IFN α , showed in the absence of stimulation an association between monocytes and plasma IFN α levels (Fig. 4d). Following R848 stimulation, both pDCs and monocytes showed an association with secreted IFN α (Fig 4h), although the percentage of IFN α + cells was lower in critical patients compared to severe and healthy controls (Fig. 4i)." (Fig. 4d actually shows the % of IFN α + pDCs, Fig. 4h shows only data for pDC: R848...)

"For this, we performed the same standardized whole blood ex vivo stimulation with recombinant IFN α 2 and measured gene expression by Nanostring as previously described."  please state where previously described

Fig S3 legend: CXCL10 (c), IL-10 (c) → CXCL10 (c), IL-10 (d)

Fig. 4c: No increase in the % of pIRF3+ monocytes was observed with the healthy control cells when treated with R848. Could the authors comment on this?

Table S2: "Débit 02 max": please translate to English

Reviewer #2 (Remarks to the Author):

The role of type I IFNs in protecting against or aggravating disease outcomes during SARS-CoV-2 infection remains controversial, and likely depends on the kinetics of IFN treatment vs. disease progression. This study adds important insights into the biological role of IFNs during different severities of COVID-19. The authors demonstrate that patients with critical COVID-19 have a more inflammatory gene expression profile. This has been demonstrated by other studies previously. However, interestingly, blood cells from hospitalized patients also mounted an inflammatory response on stimulation with TLR ligands, in contrast to cells from healthy individuals that mount an IFN-dependent ISG response. Overall, this study adds more information to the slowly growing body of literature that supports the likely role of a perturbed IFN response during severe COVID-19. Some comments for the authors are mentioned below:

1. What was the clinical criteria/score used to define patients as mild, moderate, or severe? Please elaborate on this in the methodology. While the authors have included this in the methods, it is important to note how these vary in comparison to other studies. How is moderate different from 'mild' as defined by other publications? Putting this in context is important for the readers and for reproducibility.
2. Were the patient cohorts scanned for comorbidities or ongoing treatments?
3. In Fig 1, the authors demonstrate that digital ELISA is more sensitive than other methods, such as Luminex. However, are such low levels, as detected by digital ELISA have any physiological relevance? Do these low IFN α 2 concentrations perform well (induce gene expression) when used to stimulate human cells experimentally?
4. Are the differences in Fig. 3a statistically significant?
5. Although TLR4 and TLR8 gene counts are up in hospitalized patients (Fig. 3d), why don't their cells respond to stimulation by LPS and R848 (Figs. 3a-c)? Do TLR gene counts correlate with protein levels of TLRs in the cells?
6. Label for Fig. 6f is missing
7. Heatmap legend is missing for Fig. 6b
8. MDA5 plays a major role in recognizing SARS-CoV-2 RNA. Is there a reason why MDA5 stimulation wasn't included/discussed?

20/09/2022

Dear Nature Communications

We are grateful for the careful reviews of our manuscript, which we have fully addressed as described below in a point by point response. We believe that the modifications we have made to our revised manuscript have furthered strengthened it and made our message clearer.

We look forward to your responses.

Regards
Darragh Duffy

REVIEWER COMMENTS

Reviewer #1 (Remarks to the Author):

In this manuscript, Smith, Poss  m   and Bondet et al studied type I interferon responses in samples from several COVID-19 patient cohorts in depth. First, the authors confirmed their previous findings (PMID: 32661059) showing that IFN responses were impaired in critical patients, where the concentration of IFN-alpha found in plasma samples inversely correlated with disease severity. For this, they highlighted the need to use digital ELISA, a highly sensitive assay to measure interferon concentration, as Luminex couldn't pick differences between patient groups. Moreover, and contrary to the Luminex data, data from digital ELISA correlated well with the "ISG score" (based on the relative quantification of 6 prototype ISG expression) and interferon activity (as measured by protection against virus-induced cytopathic effects). They further showed that blood cells from critical patients globally responded less well to various stimuli (poly(I:C), LPS, R848) and confirmed that their pDC count was significantly lower, whereas monocyte counts were increased. Using the nanostring technology on 800 immunology and host response related genes on patient cells exposed or not to IFN-alpha, poly(I:C) and R848, they showed a perturbation at baseline of immune responses and a lack of ISG response following stimulation in COVID-19 patients. Moreover, they could detect increased inflammatory responses in hospitalized patients. This is a highly interesting area of research and the manuscript presents an impressive amount of data. However, in order to improve the manuscript and facilitate its reading, some clarifications in the text are required. Moreover, some of the data sets used to generate some of the main figures should be shown in the supplemental figures. Please find these recommendations below.

- **We thank the reviewer for the concise summary of our study and noting its high interest and impressive amount of data. We have responded to each specific point below and made corresponding modifications to the revised manuscript.**

For clarity, in the Results section and in the figure legends, when mentioning the cohorts, the authors should always refer to the corresponding supplementary tables (this was done only for Table S3 and S5).

- **We have made this correction throughout the manuscript.**

The authors took into account the sex and age of patients in their statistical analyses. However, the sampling time (number of days post symptoms), when available, did not seem to have been taken into account in the analyses for Cohort 1 (Fig. 1, S1). Importantly, for this cohort (Table S2) the average of days post symptoms were different between the 3 categories (moderate, severe, critical) (see review figure - Days post symptom Cohort 1 and 3), and although the differences appeared not significant (using a Mann-Whitney test), it might somehow impact the findings. The authors should acknowledge this. Could the authors plot the IFN concentrations according to the number of days post symptoms for each sample (and using the disease severity color code)? This might help discard the possibility that the time of sampling parameter impacted the data.

- We thank the reviewer for highlighting this important point. We have acknowledged this in the description of the results and plotted the data as suggested, shown below for cohort 1 (Table S2), and also include in the revised supplemental Fig S1d, e & f. Analysis of which does not suggest any significant impact of days since symptoms on IFNa levels in this cohort. We had previously performed this analysis for cohort 3 (Table S4) which is included in Fig 2e and 2f.

In the third cohort (Table S4), according to date of sample and date of onset of symptoms, some samples were harvested >60 days post-symptom, in particular for patients with moderate disease (see review figure - Days post symptom Cohort 1 and 3). One might wonder why the authors kept such outliers for moderate disease and whether they impacted the results.

- We apologize for an error which caused this confusion. Within this original cohort there were some convalescent samples of the same patients after they had cleared the virus. In the final submitted manuscript we did not include analysis of these samples in order to keep a focused message on the IFN response during acute infection. However, some of these samples were listed by mistake in the supplemental table S4 which explains the observation of the reviewer that some samples were harvested >60 days post-symptom. This has now been corrected and we thank the reviewer for spotting this error.

The discrepancy between Luminex and digital ELISA data is worrying and, as such, an important finding as it could explain some of the discrepancies found in the literature. In Fig 1c (Luminex analysis) compared to 1a – b (digital ELISA) (data obtained with Cohort 1), the concentrations measured were completely different; between 10^0 to 10^3 pg/mL for instance in the healthy group with Luminex and between 10⁻¹ and 10⁻² pg/mL with digital ELISA. How could this be explained? As the values are quite high with the Luminex analyses, a problem with detection limit is not what first comes to mind. What was the limit of detection for the Luminex assay?

- We completely agree with the reviewer that the discrepancy between these technologies is worrying. We believe that it could be due to a combination of low affinity antibodies, a relatively high LOD, and a highly multiplexed assay (44 analytes) that may report non-specific background fluorescence as a low but positive signal. The LOD from the commercial supplier (Biotechne) is reported

to be 0.29pg/mL which with a dilution factor of 7 used in our analysis corresponds to 2-3 pg/mL. However, the lowest point on the standard curve is 9.8 pg/mL indicating that these low values are extrapolated from the curve. This detection limit was determined by adding two standard deviations to the mean response of twenty zero standard replicates and calculating the corresponding concentration. For the digital ELISA IFN α 2 and IFN α multi-subtype assays, the detection limits are 2fg/mL and 0.6fg/mL respectively, plus integration of the sample dilution factor. These detection limits were determined from the background level of each assay + 2SD (95% confidence) for the IFN α 2 assay and 3SD (99% confidence) for the IFN α multi-subtype assay. In Figures 1a and 1b, the concentrations we measured using the digital ELISA assays are between 10^{-2} and 10^2 pg/mL and in Fig 1c between 10^0 and 10^3 pg/mL using the Luminex assay. All the concentrations that are below the Luminex and above the digital ELISA detection limits are correctly quantified (they correlate with ISG score and IFN activity) using the digital ELISA assays, but quantified close to the detection limit by this Luminex assay, thus generating the results obtained with this test: all data is above 2-3 pg/mL, thus abolishing the differences between healthy and COVID-19 patients and between the severity classes, and correlations with ISG score and IFN activity are lost.

We have reanalyzed some of our whole blood stimulation results where there are high concentrations of IFN α after stimulation. Correlation analysis with the Simoa assays also suggests that there is a problem with calibration standards of the Luminex kit for IFN α , as there is almost 2 log differences between the reported concentrations (Figure shown below for reviewer). Despite this discrepancy, the Luminex results only correlate with the Simoa values after the strongest stimulation (R848) where the concentrations are above 10pg/mL (according to the Simoa measures). This result strongly suggests that Luminex assays should only be used to quantify high concentrations of IFN α .

Figure for reviewer. Comparison of Simoa assays with Luminex in TruCulture stimulations; Null, Poly:IC, LPS, R848. Spearman correlations were performed per stimuli on each assay comparison.

The experiments performed in Fig. 1 need to be better explained and the data provided. Notably, the ISG score and functional cytopathic experiments were not explained. For this, an article (PMID: 32661059) was quoted, but one has to download the supplemental methods from that paper to get the information. The authors should define in detail how the ISG score was obtained and provide the actual data (in a supplemental figure) for the expression measure of the 6 chosen ISGs (data corresponding to Fig. 1d-f). Moreover, it was unclear what the calibrator sample (used for 2-DDCt calculation) was. The authors should also define how the functional cytopathic effect experiment was performed (for this, reference 2, PMID: 32661059, actually referred to a paper from 1979

without further details) and the data should be provided (i.e. data corresponding to Fig. 1g-i).

- **We have included these specific details as requested in the revised manuscript, and the data has been included in the supplemental tables.**

The ISG score measured using the nanostring technology (Fig. 2b-c) should also be explained, and the data provided. Moreover, could the authors define the paired blood samples?

- **We have included additional descriptions in the Methods of the revised manuscript and included the data in the relevant supplemental tables.**

Fig. 2c-d, S2f-g: there are no units on the y-axis and y-axis, respectively

- **This has been corrected.**

In the following paragraph, the figures are mislabeled, S2b should be replaced by S2e "... This showed low transcriptional levels of all measured IFNA subtypes as expected (Fig. S2b). Despite these low baseline levels, among the 7 IFNA subtypes examined we did observe some subtype differences, with notably higher levels of IFNA6 in both COVID-19 patient groups and IFNA1/13 and IFNA5 only in the hospitalized group (Fig. S2b). IFNA2 was notably no different between all three groups (Fig. S2e)."

- **This has been corrected.**

Fig 2c-d description in the main text: (>95 considered positive): please specify the units

- **This has been included.**

Please correct the figure numbers in the following paragraph: "Additional correlation analysis between cytometry measured intracellular IFN α , and digital ELISA measured plasma IFN α , showed in the absence of stimulation an association between monocytes and plasma IFN α levels (Fig. 4d). Following R848 stimulation, both pDCs and monocytes showed an association with secreted IFN α (Fig 4h), although the percentage of IFN α + cells was lower in critical patients compared to severe and healthy controls (Fig. 4i)." (Fig. 4d actually shows the % of IFN α + pDCs, Fig. 4h shows only data for pDC: R848...)

- **This has been corrected.**

"For this, we performed the same standardized whole blood ex vivo stimulation with recombinant IFN α 2 and measured gene expression by Nanostring as previously described."  please state where previously described

- **This has been corrected.**

Fig S3 legend: CXCL10 (c), IL-10 (c) → CXCL10 (c), IL-10 (d)

- **This has been corrected.**

Fig. 4c: No increase in the % of pIRF3+ monocytes was observed with the healthy control cells when treated with R848. Could the authors comment on this?

- **We thank the reviewer for highlighting this statement. While we did not see an increase in the % of pIRF3+ monocytes we did see an increase in MFI, we have modified the text to reflect this.**

Table S2: “Débit 02 max”: please translate to English

- **This has been corrected.**

Reviewer #2 (Remarks to the Author):

The role of type I IFNs in protecting against or aggravating disease outcomes during SARS-CoV-2 infection remains controversial, and likely depends on the kinetics of IFN treatment vs. disease progression. This study adds important insights into the biological role of IFNs during different severities of COVID-19. The authors demonstrate that patients with critical COVID-19 have a more inflammatory gene expression profile. This has been demonstrated by other studies previously. However, interestingly, blood cells from hospitalized patients also mounted an inflammatory response on stimulation with TLR ligands, in contrast to cells from healthy individuals that mount an IFN-dependent ISG response. Overall, this study adds more information to the slowly growing body of literature that supports the likely role of a perturbed IFN response during severe COVID-19. Some comments for the authors are mentioned below:

- **We thank the reviewer for recognizing how our study adds important insights into the biological role of IFNs during different severities of COVID-19. We have responded to each specific point below and made corresponding modifications to the revised manuscript.**

1. What was the clinical criteria/score used to define patients as mild, moderate, or severe? Please elaborate on this in the methodology. While the authors have included this in the methods, it is important to note how these vary in comparison to other studies. How is moderate different from ‘mild’ as defined by other publications? Putting this in context is important for the readers and for reproducibility.

- **We apologize for any confusion here. We defined the patients (as described in the methods) based on internationally accepted criteria which is the requirements for supplemental oxygen at the time of sampling. Moderate patients did not require hospitalization at any timepoint. Hospitalized patients requiring supplemental oxygen via nasal cannula (maximal supplemental oxygen flow of up to 6L/min) were considered severe, with critical disease classified as requiring more than 6L of oxygen per minute, either delivered via high-flow nasal oxygen (Airvo) or a venturi mask, a clinical definition previously defined^{3,4}. In the final version of the manuscript we decided not to distinguish mild from moderate patients as this is often based on the opinion of the clinician and is thus challenging to compare across studies. We have corrected this term in the manuscript and now all patients that did not require hospitalization or oxygen supplementation are defined as moderate.**

2. Were the patient cohorts scanned for comorbidities or ongoing treatments?

- **In some of the cohorts this was possible, but due to the incomplete nature of the data sets we were not able to integrate these factors into the analysis which is why they were not included.**

3. In Fig 1, the authors demonstrate that digital ELISA is more sensitive than other methods, such as Luminex. However, are such low levels, as detected by digital ELISA have any physiological relevance? Do these low IFN α 2 concentrations perform well (induce gene expression) when used to stimulate human cells experimentally?

- **This is a very interesting point, and one that we often discuss internally. The main set of evidence that these levels are physiological come from our multitude of published studies with these assays, in particular in cases of autoimmune disease where IFN α has been clinically implicated in the pathology and where levels of the plasma protein are within these ranges (>100fg/mL). Whether these low IFN α 2 concentrations induce gene expression is dependent on the sensitivity of the assay measuring the gene expression, which do not always match the sensitivity of these digital ELISA. Recombinant IFN is often described in units, with 1 unit/mL of interferon being the quantity necessary to produce a cytopathic effect of 50%. 1 unit/mL is estimated to be between 200-300fg/mL depending on the viral cytopathic assay utilized, which also have their limitations in terms of sensitivity. However, in previous studies we have observed that an ISG score begins to correlate with Simoa measures between 1 and 10 fg/mL IFN α in SLE and JDM patients, and that the IFN activity begins to correlate between 10 and 100 fg/mL IFN α in JDM patients (Rodero et al, JEM 2016).**

4. Are the differences in Fig. 3a statistically significant?

- **No the differences between the different patient groups per stimulation are not statistically significant.**

5. Although TLR4 and TLR8 gene counts are up in hospitalized patients (Fig. 3d), why don't their cells respond to stimulation by LPS and R848 (Figs. 3a-c)? Do TLR gene counts correlate with protein levels of TLRs in the cells?

- **This is an interesting observation for which we do not currently have an evidence based explanation, but we may hypothesis that there are intracellular perturbations downstream of TLR signaling. It is challenging to measure TLR proteins in cells and were not able to perform this particular analysis.**

6. Label for Fig. 6f is missing

- **This has been corrected.**

7. Heatmap legend is missing for Fig. 6b

- **This has been corrected.**

8. MDA5 plays a major role in recognizing SARS-CoV-2 RNA. Is there a reason why MDA5 stimulation wasn't included/discussed?

- **This initial study was started during the first wave of the pandemic when little was known about the receptors that recognize SARS-CoV-2 RNA, which is the main reason why it was not included as a stimulation. We did include analysis of the levels of IFIH1 (gene encoding MDA5 in humans) in Fig 3d.**

REVIEWERS' COMMENTS

Reviewer #1 (Remarks to the Author):

The authors have addressed my concerns.